# Sustainable EFL Blended Education in Indonesia: Practical Recommendations

Putri Gayatri [1,2], Helena Sit [2,*], Shen Chen [2] and Hui Li [3]

1 Faculty of Agriculture, Universitas Brawijaya, Malang 65145, Indonesia
2 School of Education, The University of Newcastle, Newcastle, NSW 2308, Australia
3 Shanghai Institute of Early Childhood Education, Shanghai Normal University, Shanghai 200234, China
* Correspondence: helena.sit@newcastle.edu.au

**Abstract:** The establishment of a sustainable world, ecology, and economy cannot be accomplished without the success of social relations among the world's inhabitants. In the context of globalisation, which fosters the blending of various people's characters and cultures, English as an international language plays a paramount role in sustaining human relations as a tool for negotiation; it functions as a signifier of social engagement for international collaboration. Therefore, Indonesian EFL teachers should incorporate sustainable education goals into their classes so students can actively produce and use the language for real-life problem solving. This paper aims to explore a conceptual study on sustainable development integration, utilising information and communications technology (ICT) for English language teaching and learning. Through library research, the notions of sustainability are investigated and the necessity of its integration into EFL instruction is explored. Furthermore, this study recommends ICT optimisation strategies that can be employed to promote sustainable development in Indonesian EFL classrooms. This study contributes to the theory by integrating a context-based and culturally appropriate blended framework towards sustainable EFL teaching and learning in Indonesia. The integrated framework and feasible recommendations should provide practical implications for the sustainability of blended language education practices in Indonesia and in countries/regions where there are contextual similarities.

**Keywords:** digital literacy and sustainability; sustainable education development; sustainable development of EFL; ICT and EFL; sustainability of blended language education

## 1. Introduction

Currently, the international community continues to address economic and environmental issues. Since the fast growth of the global population has hampered the food supply, fresh water, land, energy, and other resources for human life [1], there has been societal awareness that future conditions may deteriorate due to current behaviours. Therefore, in 2015, the United Nations launched the 2030 Agenda for sustainable development (SD) with 17 Sustainable Development Goals (SDGs) to encourage cooperation among all global actors in order to meet the demands of the present without jeopardising the future [2]. In addition, during the past decade, groups, activists, and researchers from all around the globe have advocated for the integration of education for sustainable development (ESD) into all fields of study [3]. The goal is to strive to create abilities that enable individuals to reflect on their own behaviours, considering their current and prospective local and global social, cultural, economic, and environmental implications [4].

However, although the concept of sustainable development has been documented for a considerable period of time, it is not a simple concept for everyone to grasp, particularly for those from developing nations where English is not the main language, such as Indonesia. Since sustainable development or education for sustainable development is frequently presented in English, it is challenging for Indonesian educators to comprehend and implement the concept. Research conducted by Amran et al. in six Indonesian

schools reveals that some teachers are unfamiliar with the idea of education for sustainable development [5]. Moreover, some other recent studies find that Indonesian prospective teachers' knowledge of education for sustainable development is limited [6,7]. Even the two official documents, the biannual ADB Sustainability report and the design document for an Australian education initiative in Indonesia, *Inovasi,* are found to lack of definition of sustainability [8].

The phenomenon of a lack of knowledge towards the notion of sustainability can certainly act as a barrier in the path to the achievement of the objective. In the context of sustainable development, language and conceptual ambiguity may reduce the chance of effective implementation and successful, sustainable outcomes from donor-funded development efforts [8]. In other words, it will be challenging for individuals who do not fully comprehend the concept to contribute to the world's sustainable development.

Accordingly, the purpose of this paper is to explore a conceptual study clarifying the concept of education for sustainable development and to discuss the significance of incorporating these concepts into English as a foreign language (EFL) instruction. Moreover, this article provides practical recommendations for implementing EFL instruction for sustainable development by incorporating Information and Communication Technology (ICT) for English teachers in Indonesia and other countries with a similar sociocultural context. To achieve this objective, library research is carried out by examining and synthetising numerous articles and research findings in the fields of sustainable development, EFL instruction, and blended education.

## 2. Notion of Education Sustainability and Its Importance to the EFL Teaching

Initially, the term sustainable development was primarily concerned with the environmental aspect. Sustainability was first mentioned in the report of the World Commission on Environment and Development in 1987. It was defined as "...development that meets the needs of the present without compromising the ability of future generations to meet their own needs" [9] (p. 6). The report stated that sustainable development incorporated two major concepts: the concept of "needs" and the concept of constraints imposed by the current state of technology and social structure on the ability of the environment to meet present and future requirements [9].

While this report becomes a crucial transitory marker that sparks an industry explosion in the fields of development and sustainability [10], people start to recognise that instead of exclusively focusing on the environment sector, there is a need to see and sustain this globe as a whole unity. Thus, in 2015, the definition of sustainable development was expanded to include respect for all life—human and non-human—and natural resources, as well as incorporating issues such as poverty reduction, gender equality, human rights, education for all, health, human security, and intercultural dialogue [8]. Moreover, sustainability encompasses not simply environmental concerns; instead, it should be weaved throughout every subject, including the education system itself, to develop a comprehensive and integrated system.

### 2.1. Understanding Education for Sustainable Development

Since UNESCO has acknowledged that education is important to the concept of sustainable development as a whole [4], the idea of education for sustainable development (ESD) is then widely introduced to the global community as tools to achieve the sustainable development goals (SDGs) [11]. Sustainable education development is defined as education that helps learners to make knowledgeable decisions and take responsibility for environmental integrity, economic viability, and a just society for present and future generations, while valuing cultural variety [12]. Furthermore, educational sustainability also commits to educating the next generation about global issues and how to make a difference within themselves [13]. Moreover, on the education landscape itself, it has been accepted that the principles of sustainable development should be taught through diverse disciplines and injected into the available subjects rather than through a new exclusive subject [14].

Therefore, in accordance with this concept, learning about sustainable development is then incorporated into various topic areas, including EFL learning.

*2.2. Importance of Integrating Sustainable Development into EFL Education*

Although the economic and environmental facets of this issue often receive more attention [15], the relevance of the social aspects must not be undervalued. Consequently, language plays an essential part in the structure of human relations in its capacity as a tool for negotiating and a marker of social participation. It is important to note that there are three fundamental parts of sustainable development: (1) enhanced human learning, (2) enhanced communication, and (3) enhanced critical thinking, which are highly influenced by the language that is employed [16]. Moreover, in addition to serving as a means of communication, language provides a window into the traditions, customs, and mental processes of its users. It serves as a connecting system that leads humans down the path of culture, where the message coding and transmission system is culturally entrenched and deeply imprinted in the minds of users of a particular language [15].

Furthermore, English as a foreign language (EFL) plays a significant role in negotiating meaning in a broader context, especially in this globalisation era which promotes the mixing of different people's characters and cultures. For example, since the worldwide propositions and rules aimed to combat the rate of global warming are provided in English, the students also do a great deal in English to transfer information among diverse ethnicities and cultures [17]. In the same vein, in the field of educational research, the capacity to interact across cultures with speakers of a second/foreign language gives access to a variety of academic materials, thereby allowing both novice and expert scholars to participate in the field's ongoing activity [18]. These facts show that the usage of English will provide its speakers with increased opportunities to grow and make a positive contribution to the sustainable development of the world. Hence, since English as a global language has become one of the most important instruments for worldwide communication, English language education programmes are actively advocating for sustainable foreign language teaching and learning, which focuses on making EFL instruction as sustainable as people expect.

Unfortunately, sustainable development in EFL teaching and learning has received less attention. Even though there has been a call for years to improve the quality of the EFL teaching and learning process and ensure its sustainability for future generations, little research has been conducted in this area. For sustainable development, language and foreign language education have received far less attention in recent literature than other learning domains [19]. For example, in Indonesia, the requirement rises as educators are not only required to promote sustainable education [20] but also urged to optimise their use of technology [21]. For this reason, because ICT in education remains a difficulty for most Indonesian educators [22], a timely and relevant research study is required to determine how educational technology in EFL instruction would be aligned with sustainable foreign language education in the context of Indonesia, especially for higher education students who will be thrust into the real world upon graduation.

## 3. Blended Learning and Sustainable EFL Education in Indonesia

As it has been discussed that incorporating sustainable development into EFL instruction is necessary, it is essential to explore what constitutes sustainable, effective language learning and teaching practices in Indonesian EFL instruction and how to evaluate the success of implementing EFL education for sustainable development.

*3.1. Reconsider Blended Learning for Sustainable Development*

Blended learning, defined as the combination of face-to-face and online instruction [23,24], not only allows students to engage in a virtual learning experience, but it may also serve as a solution to the problems of a country with a big population, such as Indonesia. Indonesia is the fourth country in the world, after China, India, and America, in terms of population [25]. While this large population creates additional obstacles for Indonesia,

such as traffic congestion, pollution, waste, and inequality, it is acknowledged that blended learning can be one alternative to effectively solve the existing challenges and support the concept of sustainable development.

In the context of the Indonesian higher education, it is interesting to note that the COVID-19 pandemic has triggered more pressure to proceed to the next level of technology usage, which is Internet-based technology [26]. The context that should be underlined here is that, along with the pressing need to take care of the world's sustainability, the Indonesian Ministry of Education and Culture is of the opinion that teaching through the use of the Internet is the most forward-thinking method that can accommodate the requirements of education in the future. Therefore, the Ministry has encouraged all Indonesian educators to implement blended learning as their teaching approach by combining their traditional face-to-face teaching with online delivery to improve the teaching quality [27]. Furthermore, the policy of incorporating blended teaching and learning is also supported by the budget allocation for information technology and infrastructure in Indonesia. From 2019 until 2022, the Indonesian government has spent up to USD 4.8 million to improve the telecommunication infrastructure [28]. This commitment has led to blended learning in EFL education for sustainable development in Indonesia, emerging as a new topic to consider in the post-covid era. In light of the fact that the Indonesian government is actively exploring the feasibility of blended learning in order to improve the overall quality of education in the country, it is imperative to reconsider whether blended learning itself supports the concept of sustainable development.

According to the education authority in Indonesia, it is expected that blended learning will increase education equity in Indonesia, especially in the higher education setting. As the World Bank mentions, one possible reason for the low quality of education in Indonesia is the unequal access to education; that is, there are still disparities [29]. Blended learning, on the other hand, which incorporates both online and traditional face-to-face instruction, is believed to be able to bridge the gap in terms of distance in Indonesia. Teachers can quickly connect with their students and educate them through technology, particularly the Internet [30]. High-speed, intelligent, and robust networks are currently available to communicate with each other [31], and they can be used to easily connect teachers and students even though they are in distant locations. It means that not only can the students access the education provided by highly qualified teachers through online learning as an addition to their classroom teaching, but also the teachers can upgrade both their knowledge and teaching skills by participating in online courses, workshops, and training. Therefore, implementing blended learning gives students who reside in more country areas the opportunity to receive an education of the same quality as those who attend school in the cities. Additionally, the endeavour to provide every student in Indonesia with an equal education is consistent with the 4th sustainable development goal [4].

After all, not only can blended learning assist the concept of sustainable development by reducing traffic congestion, pollution, and waste, as well as providing equity in education, but blended learning can also facilitate the promotion of sustainable development in EFL teaching.

### 3.2. Sustainable EFL Blended Education

As a teaching approach that allows students to experience both face-to-face and online interactions, blended learning is also assessed to facilitate sustainable development integration into EFL teaching for at least two primary rationales.

First, in terms of learning English as a foreign language, blended learning facilitates the students with authentic and up-to-date materials which can enhance their critical thinking. In the context of Indonesia, where the 2013 EFL curriculum places a high emphasis on local content, which requires the textbooks to focus on Indonesian local characteristics [32,33], EFL learning can be detrimental to students because they only learn the language separately, without learning the culture that is integrated into it, and without learning how the language should be used globally. Fortunately, on the other hand, by accessing online

instructions during blended learning, students will get additional authentic materials, besides their local textbook, that can be accessed easily online. When students engage with material that provides knowledge of global issues, it brings to their attention the role they play as participants in the society in which they reside [34]. In addition, the global materials they access will expand their knowledge and train their critical thinking to resolve the field's actual challenges. If teachers are receptive and adaptable in their approach and thought processes, language instruction for sustainable development should not become an issue [34].

Second, blended learning enables students to engage with people globally without travelling abroad. The more interactions students have, the more likely they will develop their communication and negotiation skills, which will greatly assist sustainable development. In Indonesia, English is studied as a foreign language, so students have relatively limited opportunities to practise the target language outside the classroom. Thus, allowing students to genuinely use the language with a foreign speaker online will enable them to practise English, make friends, discuss global issues, and strengthen their independent and critical thinking skills [14]. Moreover, discussing global problems with the international community will cultivate persuasive discourse, negotiation, problem-solving, and exchanging experiences and knowledge, all promoting mutual understanding, tolerance, and respect [34].

After all, in the agenda for achieving the sustainable development goals, English as a foreign language should be viewed as a tool for global communication and negotiation rather than as a means of conveying ideas exclusively in Indonesia. Therefore, an authentic and genuine communication experience should be provided to the students so that they will be able to appropriately use the target language as an international language. Moreover, the above exploration of blended learning in the Indonesian context demonstrates that blended learning can be an approach that supports both sustainable development and English language instruction itself. Thus, the Indonesian government's direction in encouraging blended learning should be supported.

### 3.3. Framework to Support Sustainable EFL Blended Education

Since it has been discussed that blended learning can promote EFL education for sustainable development, it is necessary to continue studying its ideal implementation in practice. In the context of Indonesian higher education, it is understood that the government expects university students to be globally competitive [35]. They have mandated that the objective of EFL instruction at the higher education level is to enhance students' academic and professional English proficiency [36,37]. Therefore, "almost all of the tertiary level of education treats English as a general and compulsory subject to be taught to the students across departments such as technology, engineering, nursery, medicine, economics, law, agriculture, and many others" [38] (p. 240). However, although English is a required subject, the time allocation for EFL instruction is considered inadequate. Typically, English is taught for 3 to 4 credits or approximately 3 h each week over one semester [39], using a traditional method in which teachers and students must attend class together to learn the material. Moreover, instead of using English language, the Indonesian language, as well as the local language, are frequently employed as media of instruction, which makes it even less common for students to practise English as the target language in a natural setting. Nevertheless, regardless of its challenges, according to the regulation of the Indonesian Minister of Education and Culture, number 3, year 2020, about national standard for higher education, English teachers in the university have the authority and flexibility to design their own teaching and learning strategies [40], including designing their own model of blended education to achieve the expected learning outcome.

In the same way as other teaching and learning strategies, a framework for designing the teaching and learning process should be implemented to ensure the effective application of the teaching and learning process. In this case, the blended learning development framework by Mizza and Rubio [41] is employed to guide the construction of the most

effective instructional strategies that aim to make higher education students proficient in English and preserve sustainable development. This current framework is chosen because the framework is specifically designed for blended language teaching and learning. In developing the framework, they believe that the blending process should be aligned with learning philosophies (e.g., constructivism), factors central to a positive blended learning (e.g., instruction, reflection and student autonomy), and theories of second language acquisition (SLA) (e.g., input, process, and output) [41]. Considering that SLA theory is beneficial for ESL/EFL teaching and learning [42], this theoretical framework, focusing on language teaching and learning, is then considered comprehensive and practical to follow.

As explained further by Mizza and Rubio [41], design, build, teach, and revise are the four primary components that should be considered in implementing blended learning for EFL teaching. In this context, these four elements are used to guide the process of course development in the Indonesian higher education setting. These four aspects should be considered consecutively, which means that a teacher of English who wishes to implement a blended learning strategy in the classroom must begin with design. In this first step, teachers must analyse the learning objectives and consider evaluating the program's performance from the student's perspective, i.e., whether or not the students achieve the expected learning outcome. Then, teachers must construct a sustainable EFL blended teaching and learning process, which needs them to consider which activities and technologies will be applied to students. After that, teachers are obliged to deliver high-quality instruction, which necessitates a solid understanding of how to implement the teaching and learning activities they have designed. Finally, even with extensive preparation and practice, the revise stage, in which they examine their teaching method, is required to ensure that issues during the teaching and learning process are resolved, and teaching improvements are made.

In addition, because the strategy needs to be implemented in the Indonesian higher education setting, the sociocultural context of Indonesia must be taken into account when implementing this framework to develop sustainable EFL blended education. Notably, Indonesia is a country with a large power distance and is regarded as a collectivist community (country comparison). Thus, blended EFL teaching and learning development for sustainable development should rely on these cultural dimensions.

First of all, in the "design" stage of constructing a specific teaching and learning arrangement, a teacher must establish the objective and initiate an evaluation to determine if the objective is achieved [41]. As Indonesia has a large power distance, most students tend to frequently rely on their teachers to set the aim of the study. It would be challenging to conduct a needs analysis on the students to determine the purpose of education because Indonesian students do not always know what they need [43]. Accordingly, the teacher may have a larger role in determining the objective and establishing the assessment at this stage. Further, since there is an immediate need to provide EFL education for sustainable development, it is evident that the goal of blended EFL teaching and learning can be set to produce proficient users of English who can use the target language effectively in face-to-face and online discussions about sustainable development issues. In addition, to achieve this teaching objective, it must be ensured that the type of assessment reflects critical thinking and communication skills in the target language, such as conducting case studies, presenting ideas, and debating issues in the target language.

Second, after a goal has been determined, the next stage is "build", in which teachers should develop the teaching and learning experience, including deciding the use of the technology during the course [41]. To create an effective and relevant blended learning experience, the characteristics of the students must be considered at this point. Given that Indonesian students are collectivists, it is necessary for them to learn in groups. Even while one of the goals of blended learning is to develop students' independent learning skills, this does not imply that they must constantly work independently. On the contrary, students should have the opportunity to practise the target language through authentic community collaboration. Additionally, the type of technology must be determined. In the

context of Indonesia, where students may have varying degrees of comfort with technology and the Internet, the simplest and most cost-effective technology should be selected for administration at home, while the institution should provide a more advanced learning management system (LMS).

For instance, for online study at home or outside the classroom, students may use the WhatsApp application to facilitate teaching and learning. The mobile and web-based characteristics of this application make it a potential option. It means that if students do not have laptops or computers, they can access the Internet through their mobile phones, which are more affordable and possessed by most students or their families. This application enables students and teachers to study via diverse media, including video, image, and voice notes. Through the app, all participants can also share papers and presentation slides while the teacher can manage the class by forming some chat groups for different students. Additionally, WhatsApp is the most popular application in Indonesia, with 83% of internet users using it [44]. Since most students and teachers are already familiar with the application, it will be easier for them to navigate the technical aspects of the learning, allowing them to focus more on the learning material. In contrast, a web-based learning management system is advised to give students a better online learning experience at the university, considering the institution's ability to afford more advanced technology. However, as the LMS will provide more complex features, more consideration should be given to the design and implementation of web-based teaching and learning.

Since online education was viewed as revolutionary in Indonesia, it was reasonable to allocate additional resources to it so that its integration with traditional instruction would be successful. It is essential to note that there are three features of the online LMS: design, teaching instruction, and teaching and learning activity that are identified should be consistent with the sociocultural setting of Indonesia (see Figure 1).

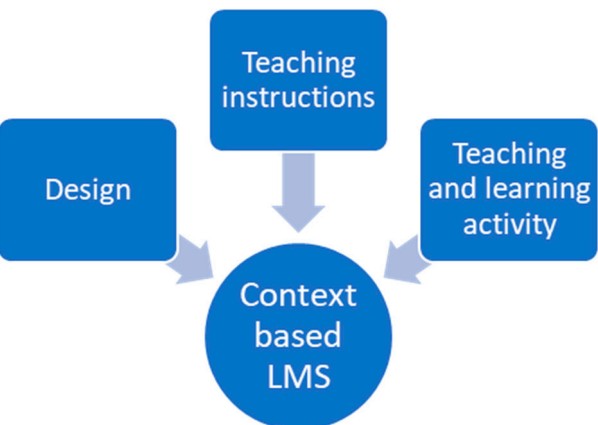

**Figure 1.** Components of online LMS and resources.

The employed online learning management system must be technologically efficient and designed with the sociocultural context of the learners in mind. Given the large power distance in Indonesia, the design development of the learning management system should be straightforward. The homepage of the website should not contain a large number of options or functions, as this would be confusing for students. Students in nations with a large power distance are not accustomed to having the flexibility to decide; hence, presenting too many features on the homepage of a website or learning management system might induce anxiety as they explore the site or system. In other words, developing a new website for students should prioritise simplicity to deliver the ideal experience in the new learning environment.

Moreover, the online environment's instruction should be consistent with the sociocultural context [45–47]. Specific and detailed instructions must be provided for Indonesia, which embraces a large power distance. Since the absence of the teacher during online learning may cause students to experience a sense of loss due to their reliance on the teacher,

the students should be provided with a detailed guide in the form of instructions in the designed learning management system. For example, the teacher should ask students to explore the LMS and instruct them on how to do so. In addition, the first page of a website with available features should provide information about those functions and indicate which ones should be examined first. The website's or LMS's detailed instructions, explanations, and stages will give clear direction and anticipate students' feelings of frustration owing to their dependability. As students gain a sense of independence, this level of specificity in the instructions should be gradually lowered. Moreover, the type of online activity on the LMS should be structured to help collectivist groups. For example, more group discussions, shared texts, and group projects should be facilitated during online instruction to allow students to remain connected to their community while learning the lesson independently.

Third, the following "teach" stage requires the teachers to determine the instructional practices [41]. Since Indonesian students embrace a large power distance, students often lack initiative. Therefore, teachers should take the initiative in the introductory course. Teachers should be able to provide clear instructions on what is expected and how to achieve it. The explicit instructions ensure that students follow the teacher's directions rather than independently determining what to do. In the case of online learning, where the teacher's involvement is minimal, the online instructions should be written properly, and the web features should be designed with a few clear feature options for operation.

Lastly, the "revise" stage should be conducted to see the effectiveness of the teaching and learning process. As with any other course, evaluating the entire teaching and learning process is vital, and can be conducted in various ways [41]. For example, to determine if the objective of instruction has been met, the student's proficiency in using the language to convey their critical thinking and problem-solving skills will be evaluated using a practice-based assessment rather than a paper-and-pencil evaluation. In addition, because a collectivist community, such as Indonesian students, may not feel comfortable directly criticising or providing feedback on their learning experience during interviews or focus group discussions, an anonymous survey might be provided to determine how students perceived the entire teaching and learning process.

## 4. Practical Recommendations for Sustainable EFL Blended Education

The preceding sections have discussed the notion of education for sustainable development, its significance to EFL teaching and learning, and the suitability of blended learning to assist EFL education for sustainable development. They have also examined the framework within which to design the approach implementation. This part concludes the paper with feasible and practical recommendations for blended EFL teaching and learning for sustainable development based on the above Indonesian-context framework.

### 4.1. Sustainable Blended Global Discussion

Sustainable blended global discussion is a blended teaching and learning approach that allows students to learn English through current global debates in both a face-to-face and online environment. The objective is for students to be able to use English to effectively communicate their perspectives, concepts, and critical thinking to resolve the global issue. During this program, in addition to practising the target language with their classmates, students can interact online with foreigners and native speakers of English. Therefore, the teacher must complete three levels of *preparation* before EFL program *implementation*, followed by an *evaluation*.

- Preparation

Stage 1: Teachers are expected to seek a global partner so that their students can engage in mutual teaching and learning. Teachers of English are strongly encouraged to establish a global network through online or in-person conferences, joint research, or simple correspondence on academic online forums such as LinkedIn or academic Facebook groups. The ability to interact, discuss, and negotiate the reciprocal online teaching and

learning process with a global partner is crucial for EFL teachers because they must be able to assemble at least a small group of students eager to join the sustainable EFL blended education together. In a case of poor fortune in which teachers are unable to find a small group of students interested in learning EFL together, they may be able to find one global partner willing to help students learn sustainable EFL by conducting a short seminar in English.

Stage 2: Teachers must ensure that the technology is operational. Students should have access to a reliable Internet connection in order to communicate with foreign speakers. In the context of education in Indonesia, where not all students may have their own device and a reliable Internet connection to begin the program, the teacher should ensure that at least one room in the institution is equipped with an LCD projector, a laptop (and speaker), and stable Internet connection to be used by a class of student. The plan is to place Indonesian students in an online group discussion with international students and provide them with communication-facilitating instructions.

Stage 3: Teachers should determine the expected learning outcome and the discussion topic; this should be relevant to sustainable development, such as climate change, poverty, geopolitical conditions, etc. The lesson's content should incorporate global education in which students are introduced to global challenges to become familiar with pertinent information and develop the skills necessary to work towards solutions [34]. Due to the fact that the teachers will pair their students with international students, the two groups of students must agree on a discussion topic. The teacher should also work closely with their online partner to plan and construct the blended global discussion program.

- Implementation

Stage 1: Teachers prepare the modification of rotation model of the blended learning by starting with the classroom teaching and learning, explaining to students the purpose of the teaching and learning process, instructing them on what to expect and what to do during class, and providing them with materials pertinent to current world issues.

Stage 2: After that, the students rotate to the initial online learning in which students are invited to search for additional information on the assigned topic on the internet. Students collaborate in small groups to discover further details and discuss the topic among themselves. Both the additional information students obtain on the Internet and their discussion must be conducted in English as the target language.

Stage 3: Once the students have understood the materials, teachers place them in an online group with other international students utilising a free video-conferencing service such as Skype or Zoom. It should be noted that education for sustainable development (ESD)-related pedagogies encourage students "to ask, analyse, think critically, and make decisions; such pedagogies move from teacher-centred to student-centred lessons and from rote memorisation to participatory learning" [48] (p.29). Thus, if the English ability of the students is insufficient, or if the technology does not support video conferencing, the online group discussion can be carried as a written chat in which the students discuss the assigned topic with the foreign speaker. Furthermore, because Indonesian students have a great deal of respect for their teachers, the teacher should be present and mediate the discussion, although the students can also play the role of mediator.

- Evaluation

After the video conference, the students may end the online discussion and share their experience and knowledge gained from this teaching and learning experience in a face-to-face environment. Small-group discussions can be used to evaluate students' knowledge, whereas an anonymous online survey can assess students' attitudes towards the program. Students will receive teachers' comments to improve their performance in this stage. Depending on the teacher's agreement with his/her global partner, this activity can be undertaken at least once a month.

Because there is often a shortage of professional training on appropriate pedagogies when education for sustainable development should be employed [49], this practical recommendation should be a feasible reference for any EFL teachers working in Indonesia.

### 4.2. Potential Challenges

Given that sustainable EFL blended education is a novel concept for Indonesian practitioners, several challenges may arise throughout its implementation. First, the practical recommendation in this study requires the teacher to have global partners, which is not always simple to implement. As described in the preparation section, this requirement can be accomplished by approaching a foreign educator who has past research collaboration or by approaching a foreign educator at an international conference attended by the teachers. Unfortunately, teachers cannot always perform this action. Not all of them may have the opportunity to attend international conferences abroad or to collaborate on research with other foreign English teachers. If teachers have a limited worldwide network, it is recommended that they participate in international online group discussions, such as joining ELT-focused Facebook pages. Once a teacher has a Facebook account, they can search for online discussion groups for EFL teachers using the keywords of ELT education or EFL education. The teacher might then examine the members' profiles to find a possible global partner. Teachers may consider the members' country of origin to accommodate the time difference. For instance, Indonesian educators may cooperate with educators from Australia, Singapore, Japan, or Malaysia, whose time difference is not too significant, so that it does not hinder the plan for having live collaboration during the program. The teacher may also consider cooperating with educators whose students have comparable proficiency levels. To find the most significant global partner to support the students, the capability to propose a program and discuss it is highly encouraged. In addition, teachers can join academic online platforms, such as ResearchGate or LinkedIn, to assess the interests of other educators and endeavour to contact individuals with a strong teaching profile and an interest in ICT in education. It should be remembered that such actions may look challenging at the beginning. However, the fact is that many educators are open to collaborating in teaching and research today, so it should not be difficult to locate at least one individual to support the program.

Second, regarding the students, it must be ensured that the activity is meaningful, accessible, and safe for all participants. Live spoken communication is encouraged at school due to the varying capabilities of students to access technology outside of the classroom, as well as to ensure the appropriateness of the students' interactions. The online interactions can be staged, beginning with listening to a short seminar discussing sustainable development and launching question-and-answer sessions. The objective is to provide Indonesian students with the opportunity to utilise English in a real-world setting and converse with foreigners during an online seminar. This activity can be modified based on each teacher's global partners. For instance, if the teacher can reach an agreement with other teachers to include more students, the students' attempts to acquire EFL can be conducted through teacher-monitored online written group discussion. Whenever possible, assuming the institution has enough laptops to accommodate the number of students, teachers may permit students to work in pairs utilising a single laptop to communicate orally with other English speakers abroad. Once students get the chance to use the target language in authentic communication, they will learn the language and how to utilise it as a negotiation tool to promote sustainable development goals.

## 5. Conclusions

This paper discusses blended learning in Indonesian higher education and considers a new approach to support and facilitate EFL teaching and learning. This study provides an understanding of blended learning and how this pedagogical approach fits sustainable EFL teaching at the tertiary level in Indonesia. This paper also presents a conceptual study on sustainable development integration utilising ICT for sustainable EFL education. It

investigates notions of sustainability and explores the necessity of its integration into EFL instruction. The integration of sustainable development as a concept to fulfil the current demands while maintaining future needs has been accepted for EFL instruction. Furthermore, since studying a foreign language is essential for effective global communication and negotiation, students should not only learn how to construct correct sentences in the target language, but also how to use the language effectively in the interest of world sustainable development.

Furthermore, it can be argued that theoretically and technologically blended learning supports EFL teaching and learning for sustainable development, so its application, together with the Indonesian sociocultural context consideration, is recommended. Moreover, the practical recommendations that Indonesian EFL educators can adopt/adapt in their sustainable EFL blended education are presented. This study makes a contribution to the theory by integrating a context-based and culturally appropriate blended framework towards sustainable EFL teaching and learning in Indonesia.

The study is limited in its study scope. Though the theoretical perspective is solid, the effectiveness of the theoretical framework needs to be further evaluated and assessed, presumably, through large-scale quantitative or qualitative research. Despite this limitation, the integrated framework and feasible recommendations in this article consider the sociocultural context of Indonesia, which should provide practical implications for the sustainability of blended language education practices in Indonesia and in countries/regions where there are contextual similarities.

**Author Contributions:** Conceptualization, P.G., H.S., S.C. and H.L.; writing—original draft preparation, P.G and H.S.; writing—review and editing, all authors; visualization, P.G. All authors have read and agreed to the published version of the manuscript.

**Funding:** This research received no external funding.

**Institutional Review Board Statement:** This is a library research study that does not need ethics application by the university and institution.

**Informed Consent Statement:** Not applicable.

**Data Availability Statement:** Not applicable.

**Acknowledgments:** The authors thank the MDPI and editor team's valuable review and constructive feedback.

**Conflicts of Interest:** The authors declare no conflict of interest.

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
