# Peer review of "Sustainable EFL Blended Education in Indonesia: Practical Recommendations"

_sustainability, doi:10.3390/su15032254_

Round 1

Reviewer 1 Report

1.     In the Abstract, the authors have not mentioned the purpose of the study. In addition, there is no brief note on the methodology adopted for the study, and there is no nutshell on the results obtained. These are all crucial to mention in the abstract. But they are missing.

2.     What are the variables used in this study? They are not clearly mentioned.

3.     There is no literature review section/research questions/hypotheses. How did the authors develop the study without having all these important? The papers need more detailed analysis in this case.

4.     What type of methodology was adopted in this paper to excute the research? It was not stated by the authors.

5.     The paper seems like a mere summary of the existing literature.

6.     What are the results or findings of the study? How did the authors conclude the research? These details are not elaborated properly.

7.     How do the authors give Practical Recommendations for Sustainable EFL Blended Education? Whether they use any variables or methodology to justify the same?

8.     What are the limitations of the study?

9.     There is no discussion to discuss the present study results with existing study results.

10.  The paper needs a lot of improvements to consider for publication. This form of the paper lacks most of the important elements.

Author Response

Response to Reviewer 1 Comments

Point 1:   In the Abstract, the authors have not mentioned the purpose of the study. In addition, there is no brief note on the methodology adopted for the study, and there is no nutshell on the results obtained. These are all crucial to mention in the abstract. But they are missing.

 Response 1: Thank you very much for your feedback. We apologise if the purpose of the study, the methods, and the findings are considered unclear in the abstract. We have revised our abstract by stressing the paper's objectives and identifying library research as the method through which we convey the conceptual investigation in this article. In addition, since we do not conduct empirical research, we believe the remaining sentences should represent the manuscript's content.

“This paper aims to explore a conceptual study on sustainable development integration utilising information and communications technology (ICT) for English language teaching and learning. Through library research, the notions of sustainability are investigated and the necessity of its integration into EFL instruction is explored. Furthermore, this study recommends the ICT optimisation strategies that can be employed to promote sustainable development in Indonesian EFL classrooms. This study contributes to the theory by integrating a context-based and culturally appropriate blended framework towards sustainable EFL teaching and learning in Indonesia. The integrated framework and feasible recommendations should provide practical implications for the sustainability of blended language education practices in Indonesia and in countries/regions where there are contextual similarities”

Point 2: What are the variables used in this study? They are not clearly mentioned.

Response 2: Thank you for bringing up this question. We have modified the article to include the study methodologies and variables at the end of the introduction section.

To achieve this objective, library research is carried out by examining and synthetising numerous articles and research findings in the fields of sustainable development, EFL instruction, and blended education” (line 71-73)

Point 3: There is no literature review section/research questions/hypotheses. How did the authors develop the study without having all these important? The papers need more detailed analysis in this case.

Response 3: Thank you very much for your questions. The article contains both the literature review and research questions. However, due to the nature of library research, the article's organisation may differ from that of the empirical study. In our research, instead of posing questions in the form of interrogative sentences, we utilise affirmative sentences to demonstrate the purpose of the study, which is:

The purpose of this paper is to explore a conceptual study clarifying the concept of education for sustainable development and to discuss the significance of incorporating these concepts into English as a foreign language (EFL) instruction. Besides, this article provides practical recommendations for implementing EFL instruction for sustainable development by incorporating Information and Communication Technology (ICT) for English teachers in Indonesia and other countries with a similar sociocultural context” (in line 65f).

In addition, this objective was established as a result of a literature review that revealed a gap between the requirement to convey sustainable EFL learning and the lack of understanding regarding the sustainability concept.

Point 4: What type of methodology was adopted in this paper to execute the research? It was not stated by the authors.

Response 4: Thank you very much for your question. This article employs library research to present the conceptual study on the sustainable EFL blended teaching in Indonesia. As we respond to your second feedback, we also have addressed this issue in lines 71-73.

To achieve this objective, library research is carried out by examining and synthetising numerous articles and research findings in the fields of sustainable development, EFL instruction, and blended education”

Point 5:  The paper seems like a mere summary of the existing literature.

Response 5: We appreciate your comment. Since we undertake a conceptual study, the structure of our article differs from that of empirical research. Therefore, it may appear to be a summary of prior existing literature. However, we believe that our study has a solid foundation, as we have identified the gap between the necessity to execute sustainable EFL teaching in Indonesia and the lack of comprehension of the idea. In addition, we have studied a significant amount of literature, examining the history of the notion, looking at the recent research findings in the area, and even considering the sociocultural context of Indonesia. Since the latest 2013 curriculum-revision was authorized by the Indonesian government that highly promotes and encourages the professional development of digital literary, and since blended teaching and learning across the curriculum is highly encouraged, few research is available in the Indonesian higher education sector related to sustainable blended EFL education. Therefore, this current article should provide EFL teachers in Indonesian higher education with a timely, useful instructional reference/guideline on how to establish sustainable EFL blended education.

Point 6:  What are the results or findings of the study? How did the authors conclude the research? These details are not elaborated properly.

Response 6: Thank you very much for the question. The result of the study is the clarification on the concept of sustainable development as a result of an extensive and comprehensive review study on various literature related to the topic. Also, part of the result of the study is the insight into why and how sustainable development should be integrated into blended EFL education in Indonesia. With the same nature, investigation and exploration of why and how sustainable blended EFL teaching is implemented are conducted through examining various literature on the related topic, highlighting the significant points and considering the cultural dimensions of Indonesia to propose the practical recommendation for Indonesian EFL teachers. The examination and exploration in the article are written as a comprehensive unit, not in a separate section, as appeared in the organisation of most empirical research.

Point 7:  How do the authors give Practical Recommendations for Sustainable EFL Blended Education? Whether they use any variables or methodology to justify the same?

Response 7: Thank you very much for raising this issue. This paper makes practical recommendations based on the extensive and comprehensive review and discussion on the sustainable blended EFL framework. Since the article examines the EFL framework and how it can be adapted in regard to sustainable development and the sociocultural context of Indonesia, the result of this discussion is then used to propose practical recommendations that can serve as an example and instructional guide to provide EFL practitioners in Indonesian higher education with a clearer picture on how to implement the approach.

Point 8:  What are the limitations of the study?

Response 8: Thank you very much for pointing this aspect on our study. We have revised our paper and included the limitation and future directions of the study in the conclusion part.

“The study is limited in its study scope. Though the theoretical perspective is solid, the effectiveness of the theoretical framework needs to be further evaluated and assessed, presumably, through a large scare of quantitative or qualitative research. Despite the limitation, the integrated framework and feasible recommendations in this article considers the sociocultural context of Indonesia, which should provide practical implications for the sustainability of blended language education practices in Indonesia and in countries/regions where there are contextual similarities“ (In line 638-644).

Point 9:  There is no discussion to discuss the present study results with existing study results.

Response 9: Thank you so much for your feedback. Due to the nature of conceptual and library study, rather than having a specific section which generally presents discussions of results with existing study result, we include our review results, analysis, synthesis, and discussions throughout the paper. Hence, this study can make a contribution to the theory by integrating a context-based and culturally appropriate blended framework towards sustainable EFL teaching and learning in Indonesia.

Point 10:  The paper needs a lot of improvements to consider for publication. This form of the paper lacks most of the important elements.

Response 10: We appreciate your constructive feedback. Based on your and the other four reviewers' comments, we have carefully followed your comments and revised the article accordingly to enhance its quality for publication. We recognise that, as a perspective-type study that does not incorporate empirical research, our paper appears to lack some features. Some elements such as hypothesis or research findings may not be specifically mentioned in this paper due to its different research types. As the other four reviewers have noticed, several elements of the review are not applicable to this manuscript due to its nature. However, although this article is not based on empirical research, other review reports agree that we provide a good foundation for studying the case and have thoroughly analysed the topic to make the best possible practical proposal for a sustainable EFL blended education in Indonesia.

Once again, we highly value all of the constructive feedback on our manuscript. If additional modifications or information are required, we are happy to consider and make revisions. Thank you once more for all the positive feedback and helpful suggestions for enhancing the quality of this manuscript.

Best regards,

The authors

Reviewer 2 Report

This is a valid and interesting study which makes a contribution to scholarship. It is well-written, using appropriate academic style and structure. It is highly relevant to the Journal and includes relevant background information about Sustainable Development Goals. Hence this review is generally positive, with just a few matters for the authors to consider:

1.    Title: certainly it is not a case study – but does this need to be part of the title? Perhaps revise the title and stress that it is mainly conceptual with some more practical recommendations.

2.    Whilst the main focus is on English (EFL) at tertiary level in Indonesia, and this is made clear, perhaps it is not made clearly enough throughout the paper text. There is no reference at all to the languages (mainly but not only Bahasa Indonesia) that the students and their instructors bring to the EFL classes, whether face-to-face physical, blended, or fully online. This seems an odd omission when the conclusion highlights the importance of sociocultural context.

3.    There should be more analysis and discussion of the role of English in tertiary-level education in Indonesia: reading, EAP, academic literacy skills including online digital literacies, e.g. smart searching using Google Scholar for references.

4.    There is not enough critical engagement with the notion of blended learning. Uncritical adoption of blended learning, mandated by Ministries of Education (line 136) reflects a general international trend since the Covid19 pandemic forced all learning and teaching to migrate to online modes, but it is certainly not a panacea. Even if it is able to bridge gaps (digital divide, line 173), there should be consideration of how this might happen, including through the use of smartphones (which most students have) as opposed to I-pads or laptops which many in rural areas may not have.

(From here onwards, minor matters noted by the reviewer while reading through the manuscript:)

5.    Line 118: the phrase – maybe a citation – “among distinct crossroads” seems odd. Otherwise control of micro-level language, structure and style is admirable.

6.    Valid point about the lack of attention to EFL issues and SDGs/sustainable development (line 125): this creates a research space and rationale for the paper.

7.    Valid points are made about the need for a learning philosophy and awareness of  theories of Second Language Acquisition (line 241 f; but is it really SLA in the Indonesian context? FLA?). With reference to the model adapted from Miller and Rubio, one might question the ‘design’ component: to what extent are EFL instructors free to design their own curriculum? Usually this is imposed from above, or from coursebooks.

8.    There is good discussion of the roles of Learning Management Systems (line 284f): navigating these requires high levels of digital literacy by students – and maybe initially require specific instruction at the start of a module or course.

9.    Line 345: the “large power distance” found in Indonesia is repeated from line 334 – could be rephrased with a synonym, maybe.

Author Response

Response to Reviewer 2 Comments

Point 1: This is a valid and interesting study which makes a contribution to scholarship. It is well-written, using appropriate academic style and structure. It is highly relevant to the Journal and includes relevant background information about Sustainable Development Goals. Hence this review is generally positive, with just a few matters for the authors to consider:

Title: certainly it is not a case study – but does this need to be part of the title? Perhaps revise the title and stress that it is mainly conceptual with some more practical recommendations.

Response 1: Thank you very much for recognising this paper and for the constructive feedback that will help to improve its quality. Regarding the title, the authors agree to modify it so that it more accurately reflects the article's content. The title is revised as: Sustainable EFL Blended Education in Indonesia: Practical Recommendations.

Point 2: Whilst the main focus is on English (EFL) at the tertiary level in Indonesia, and this is made clear, perhaps it is not made clearly enough throughout the paper text. There is no reference at all to the languages (mainly but not only Bahasa Indonesia) that the students and their instructors bring to the EFL classes, whether face-to-face physical, blended or fully online. This seems an odd omission when the conclusion highlights the importance of sociocultural context.

Response 2: Thank you for bringing this to our attention. The authors agree to make the EFL teaching context in Indonesia clearer. The authors have therefore added a paragraph explaining the EFL teaching setting, including the language that students and teachers typically employ during EFL teaching and learning (line 269-272). 

“Besides, instead of using English language, the Indonesian language as well as the local language are frequently employed as a medium of instruction; making it even less common for students to practise English as the target language in a natural setting.”

Point 3: There should be more analysis and discussion of the role of English in tertiary-level education in Indonesia: reading, EAP, academic literacy skills including online digital literacies, e.g. smart searching using Google Scholar for references.

Response 3: Thank you very much for pointing this issue. The authors have written additional information related to EFL teaching context in Indonesia, including the aim of EFL teaching in Indonesian higher education (line 259-264).

“In the context of Indonesian higher education, it is understood that the government expects university students to be globally competitive [35]. They have mandated that the objective of EFL instruction at the higher education level is to enhance students' academic and professional English proficiency [36, 37]. Therefore, "almost all of the tertiary level of education treats English as a general and compulsory subject to be taught to the students across departments such as technology, engineering, nursery, medicine, economics, law, agriculture, and many others" [38] (p.240)”

Point 4: There is not enough critical engagement with the notion of blended learning. Uncritical adoption of blended learning, mandated by Ministries of Education (line 136) reflects a general international trend since the Covid19 pandemic forced all learning and teaching to migrate to online modes, but it is certainly not a panacea. Even if it is able to bridge gaps (digital divide, line 173), there should be consideration of how this might happen, including through the use of smartphones (which most students have) as opposed to I-pads or laptops which many in rural areas may not have.

Response 4: Thank you very much for your thoughtful feedback in this area. The authors agree that more discussion on how blended learning may happen (including through the use of smartphones) should be provided in the paper to support the bridge gaps that is already mentioned. Thus, in the framework development section, the authors have provided example on how blended learning may be accommodated outside the classroom by using mobile phone and affordable application (line 347f).

“For instance, for online study at home or outside the classroom, students may use the WhatsApp application to facilitate teaching and learning. The mobile and web-based characteristics of this application make it a potential option. It means that if students do not have laptops or computers, they can access the Internet through their mobile phones, which are more affordable and possessed by most students or their families. This application enables students and teachers to study via diverse media, including video, image, and voice notes. Through the app, all participants can also share papers and presentation slides while the teacher can manage the class by forming some chat groups for different students. Additionally, WhatsApp is the most popular application in Indonesia, with 83% of internet users using it [44]. Since most students and teachers are already familiar with the application, it will be easier for them to navigate the technical aspects of the learning, allowing them to focus more on the learning material. In contrast, a web-based learning management system is advised to give students a better online learning experience at the university, considering the institution's ability to afford more advanced technology. However, as the LMS will provide more complex features, more consideration should be given to the design and implementation of web-based teaching and learning.”

Point 5: Line 118: the phrase – maybe a citation – “among distinct crossroads” seems odd. Otherwise control of micro-level language, structure and style is admirable.

Response 5: Thank you very much for your detailed comments. The authors have paraphrased the citation to make it clearer and simpler to comprehend (currently in line 129-132).

“since the worldwide propositions and rules aimed to combat the rate of global warming are provided in English, the students also do a great deal in English to transfer information among diverse cross-ethnic and culture [17]”

Point 6: Valid point about the lack of attention to EFL issues and SDGs/sustainable development (line 125): this creates a research space and rationale for the paper.

Response 6: We appreciate your positive review of the article.

Point 7:   Valid points are made about the need for a learning philosophy and awareness of theories of Second Language Acquisition (line 241 f; but is it really SLA in the Indonesian context? FLA?). With reference to the model adapted from Miller and Rubio, one might question the ‘design’ component: to what extent are EFL instructors free to design their own curriculum? Usually this is imposed from above, or from coursebooks.

Response 7: Thank you very much for your recognition to the needs of learning philosophy and thank you for pointing this crucial issue. The authors agree that in the Indonesian context, English is taught as foreign language instead of second language. However, we believe that SLA theory is also relevant to support the EFL teaching and learning. We have put a citation to support our argument that SLA is relevant to be incorporated in the EFL setting (line 287).

“Considering that SLA theory is beneficial for ESL/EFL teaching and learning [42], this theoretical framework focusing on language teaching and learning is then considered comprehensive and practical to follow”

Also, related to the curriculum design, we have provided more information on the government’s regulation, stating that English teachers in the Indonesian university have the flexibility and authority to design their own teaching strategies in order to achieve the aim of teaching and learning process (line 272-276)

“Nevertheless, regardless of its challenges, according to the regulation of the Indonesian Minister of Education and Culture number 3 year 2020 about national standard for higher education, English teachers in the university have the authority and flexibility to design their own teaching and learning strategies [40], including designing their own model of blended education to achieve the expected learning outcome.”

Point 8:   There is good discussion of the roles of Learning Management Systems (line 284f): navigating these requires high levels of digital literacy by students – and maybe initially require specific instruction at the start of a module or course.

Response 8: Thank you for pointing this aspect. In accordance with the recommendations of other reviewers, the authors have attempted to clarify this section by providing examples and rearranging the paragraphs to demonstrate how mobile and web-based learning using LMS can be adapted to the context of Indonesia, including the explanation that specific instructions are required in the LMS at the beginning of an online learning program (in line 347-410)

Point 9:    Line 345: the “large power distance” found in Indonesia is repeated from line 334 – could be rephrased with a synonym, maybe.

Response 9: We value your recommendation. However, we would prefer to maintain the term of "large power distance" for consistency and to make the Indonesian cultural dimension evident to the reader.

Again, we highly value all of the constructive feedback on our manuscript. If additional modifications or information are required, we are happy to consider and make revisions. Thank you once more for all the positive feedback and helpful suggestions for enhancing the quality of this manuscript.

Best regards,

The authors

Reviewer 3 Report

You are addressing an important and timely topic. However, some issues need to be addressed to improve the quality of the manuscript. Below I describe some of the issues.

Title: Suggest adding "blended" to the title, as this is the paper's focus.

Educational Context: The authors provide helpful information about the Indonesian context. However, more information about the educational context in Indonesia is needed. For example, is this framework for English teaching in primary schools? Secondary schools? Language schools? Higher education? You need to specify the particular educational context in the paper.

EFL Teaching in Indonesia: It would be nice to have more information about EFL teaching in Indonesia. How many hours of English instruction a week do institutions offer? What type of EFL instruction is usually offered? What instructional materials are usually used for English teaching? What are the significant needs of teachers and learners?

Blended Education: The authors need to specify what type of blended education they propose for implementation in the Indonesian context. Which type of blended education would be more effective in the Indonesian EFL context? Is it a rotation model? Flex model? Enriched model? The particular blended education model needs to be specified in the paper.

Access to Technology: The authors discuss the need for improving connectivity issues (i.e., access to high-speed internet). However, there needs to be more discussion about EFL learners in Indonesia having access to technological devices (e.g., computers, laptops, tablets, smartphones) outside the classroom.

Recommendations: I wonder if some of the author's recommendations are feasible. For example, can all teachers have global partners? Can all students have access to technology at home? Can students interact with native English speakers in schools when using one computer? Can students communicate with students in English-speaking countries, given time differences? I suggest adding a section on potential challenges for implementing the blended education framework in Indonesia and potential solutions for overcoming them.

Other Issues. The authors mentioned the need for teacher education and training as an impediment to implementing this framework in Indonesia. What recommendations can you provide to solve this problem? How can teachers overcome this problem?

Author Response

Response to Reviewer 3 Comments

Point 1: You are addressing an important and timely topic. However, some issues need to be addressed to improve the quality of the manuscript. Below I describe some of the issues.

Title: Suggest adding "blended" to the title, as this is the paper's focus.

Response 1: Thank you very much for your appreciation and meaningful feedback. We agree with your suggestion and have revised the title into: Sustainable EFL Blended Education in Indonesia: Practical Recommendations.

Point 2: Educational Context: The authors provide helpful information about the Indonesian context. However, more information about the educational context in Indonesia is needed. For example, is this framework for English teaching in primary schools? Secondary schools? Language schools? Higher education? You need to specify the particular educational context in the paper.

Response 2: Thank you very much for bringing up this issue. The authors have made revision in some parts, such as in line 152, 190, 281, 296,and 314 clarifying if the framework here is adapted to provide practical recommendation for Indonesian higher education context.

Point 3: EFL Teaching in Indonesia: It would be nice to have more information about EFL teaching in Indonesia. How many hours of English instruction a week do institutions offer? What type of EFL instruction is usually offered? What instructional materials are usually used for English teaching? What are the significant needs of teachers and learners?

Response 3: Thank you very much for your suggestion. We agree with your suggestion and we have put additional information in line 259-276 related to EFL teaching and learning in the Indonesian higher education context.

“In the context of Indonesian higher education, it is understood that the government expects university students to be globally competitive [35]. They have mandated that the objective of EFL instruction at the higher education level is to enhance students' academic and professional English proficiency [36, 37]. Therefore, "almost all of the tertiary level of education treats English as a general and compulsory subject to be taught to the students across departments such as technology, engineering, nursery, medicine, economics, law, agriculture, and many others" [38] (p.240). However, although English is a required subject, the time allocation for EFL instruction is considered inadequate. Typically, English is taught for 3 to 4 credits or approximately 3 hours each week over one semester [39] using a traditional method in which teachers and students must attend class together to learn the material. Besides, instead of using English language, the Indonesian language as well as the local language are frequently employed as a medium of instruction; making it even less common for students to practise English as the target language in a natural setting. Nevertheless, regardless of its challenges, according to the regulation of the Indonesian Minister of Education and Culture number 3 year 2020 about national standard for higher education, English teachers in the university have the authority and flexibility to design their own teaching and learning strategies [40], including designing their own model of blended education to achieve the expected learning outcome.”

Point 4: Blended Education: The authors need to specify what type of blended education they propose for implementation in the Indonesian context. Which type of blended education would be more effective in the Indonesian EFL context? Is it a rotation model? Flex model? Enriched model? The particular blended education model needs to be specified in the paper.

Response 4: Thank you very much for bringing up this to our attention. The authors previously did not mention specifically the type of blended learning model in this article because it is found that the proposed blended education in this article cannot be strictly classified as Rotation, Flex, A La Carte, or Enriched model due to the incorporation of socio-cultural context of Indonesia. However, to give clearer picture to the reader on the practical recommendation we propose, we have revised the manuscript by mentioning in line 536-539 that the blended education here is “the modification of the rotation model” due to similar nature of its implementation with the rotation model.

“Stage 1: Teachers prepare the modification of rotation blended model by starting with the classroom teaching and learning, explaining to students the purpose of the teaching and learning process, instruct them on what to expect and what to do during class, and provide them with materials pertinent to current world issues.”

Point 5: Access to Technology: The authors discuss the need for improving connectivity issues (i.e., access to high-speed internet). However, there needs to be more discussion about EFL learners in Indonesia having access to technological devices (e.g., computers, laptops, tablets, smartphones) outside the classroom.

Response 5: Thank you very much for your insight on the technological access for students outside the classroom. The authors have provided more discussion by giving an example on the type of technology devices and software to be used outside the class by the students (line 347f).

“For instance, for online study at home or outside the classroom, students may use the WhatsApp application to facilitate teaching and learning. The mobile and web-based characteristics of this application make it a potential option. It means that if students do not have laptops or computers, they can access the Internet through their mobile phones, which are more affordable and possessed by most students or their families. This application enables students and teachers to study via diverse media, including video, image, and voice notes. Through the app, all participants can also share papers and presentation slides while the teacher can manage the class by forming some chat groups for different students. Additionally, WhatsApp is the most popular application in Indonesia, with 83% of internet users using it [44]. Since most students and teachers are already familiar with the application, it will be easier for them to navigate the technical aspects of the learning, allowing them to focus more on the learning material. In contrast, a web-based learning management system is advised to give students a better online learning experience at the university, considering the institution's ability to afford more advanced technology. However, as the LMS will provide more complex features, more consideration should be given to the design and implementation of web-based teaching and learning.”

However, the authors intend to provide practical implementation for any English teachers in Indonesian universities regardless of their capacity in ICT. Therefore, in the section under "Practical Recommendations," the authors advocate that online learning during blended learning be conducted at school with the least advantageous technology access that students and teachers may have. In line 521-523, the authors believe that anyone can execute this advice since “the teacher should ensure that at least one room in the institution is equipped with an LCD projector, a laptop (and speaker), and stable Internet connection to be used by a class of student”

Point 6: Recommendations: I wonder if some of the author's recommendations are feasible. For example, can all teachers have global partners? Can all students have access to technology at home? Can students interact with native English speakers in schools when using one computer? Can students communicate with students in English-speaking countries, given time differences? I suggest adding a section on potential challenges for implementing the blended education framework in Indonesia and potential solutions for overcoming them.

Response 6: Thank you very much for your thought-provoking recommendation. The authors agree and have included one additional section regarding "potential challenges" before the conclusion. The authors added two paragraphs to this section to address potential obstacles and potential solutions that may appear during the program (in line 569f).

“          Given that sustainable EFL blended education is a novel concept for Indonesian practitioners, several challenges may arise throughout its implementation. First, the practical recommendation in this study requires the teacher to have global partners, which is not always simple to implement. As described in the preparation section, this requirement can be accomplished by approaching a foreign educator who has past research collaboration or by approaching a foreign educator at an international conference attended by the teachers. Unfortunately, teachers cannot always perform this action. Not all of them may have the opportunity to attend international conferences abroad or to collaborate on research with other foreign English teachers. If teachers have a limited worldwide network, it is recommended that they participate in international online group discussions, such as joining ELT-focused Facebook pages. Once a teacher has a Facebook account, they can search for online discussion groups for EFL teachers using the keywords of ELT education or EFL education. The teacher might then examine the member's profile to find a possible global partner. Teachers may consider the member's country of origin to accommodate the time difference. For instance, Indonesian educators may cooperate with educators from Australia, Singapore, Japan, or Malaysia, whose time difference is not too significant so that it does not hinder the plan for having live collaboration during the program. The teacher may also consider cooperating with educators whose students have comparable proficiency levels. To find the most significant global partner to support the students, the capability to propose a program and discuss it is highly encouraged. In addition, teachers can join academic online platforms, such as ResearchGate or LinkedIn, to assess the interests of other educators and select to contact individuals with a strong teaching profile and an interest in ICT in education. It should be remembered that such actions may look challenging at the beginning. However, the fact is that many educators are open to collaborating in teaching and research today, so it should not be difficult to locate at least one individual to support the program.

Second, regarding the students, it must be ensured that the activity is meaningful, accessible, and safe for all participants. Live spoken communication is encouraged at school due to the varying capabilities of students to access technology outside of the classroom, as well as to ensure the appropriateness of the student's interaction. The online interactions can be staged, beginning with listening to a short seminar discussing sustainable development and launching question-and-answer sessions. The objective is to provide Indonesian students with the opportunity to utilise English in a real-world setting and converse with foreigners during an online seminar. This activity can be modified based on each teacher's global partners. For instance, if the teacher can reach an agreement with other teachers to include more students, the students' attempts to acquire EFL can be conducted through teacher-monitored online written group discussion. Whenever possible, assuming the institution has enough laptops to accommodate the number of students, teachers may permit students to work in pairs utilising a single laptop to communicate orally with other English speakers abroad. Once students get the chance to use the target language in authentic communication, they will learn the language and how to utilise it as a negotiation tool to promote sustainable development goals.”

Point 7: Other Issues. The authors mentioned the need for teacher education and training as an impediment to implementing this framework in Indonesia. What recommendations can you provide to solve this problem? How can teachers overcome this problem?

Response 7: Thank you for highlighting this section. In line 565-567, the authors suggest that the practical recommendation offered in this study is meant to aid teachers in navigating their sustainable EFL blended practises, since proper teacher training on the topic is occasionally lacking in Indonesia.

Again, we highly value all of the constructive feedback on our manuscript. If additional modifications or information are required, we are happy to consider and make revisions. Thank you once more for all the positive feedback and helpful suggestions for enhancing the quality of this manuscript.

Best regards,

The authors

Reviewer 4 Report

The paper offers an interesting perspective on the sustainable education development of EFL through blended teaching and learning and the use of ICT. After providing a theoretical background, the paper offers a clear framework for sustainable education development and practical recommendations for the classroom. The practical approach is closely related to Indonesia, emphasizing the specificities in EFL teaching in the country, but it can be implied in the regions with contextual similarities as well. This last aspect is very important and provides general outline for the paper. The framework and recommendations are clearly presented and thoroughly explained within the specific context, which is a great asset of the paper. The selection of references is relevant and recently published.

There are only a few segments to consider improving:

1. In lines 61-64, the authors claim: “Besides, this article provides English teachers in Indonesia with practical recommendations for implementing EFL education for sustainable development through the integration Information and Communication Technology (ICT).” Although the recommendations are beneficial to English teachers in Indonesia, here, as well as in some other places in the paper, it should be added that other contextually similar teaching environments can also benefit from it.

2. In lines 88-92, the end of citation from the reference 12 is clear, but the beginning of the citation is unknown.

3. In line 440, the tertiary level is related to the country (Indonesia) not the language (Indonesian).

4. Although the English language and grammar is correct, the paper could benefit from the proofreading by an expert, since a number of sentences do not sound as English natural.

Author Response

Response to Reviewer 4 Comments

Point 1: The paper offers an interesting perspective on the sustainable education development of EFL through blended teaching and learning and the use of ICT. After providing a theoretical background, the paper offers a clear framework for sustainable education development and practical recommendations for the classroom. The practical approach is closely related to Indonesia, emphasizing the specificities in EFL teaching in the country, but it can be implied in the regions with contextual similarities as well. This last aspect is very important and provides general outline for the paper. The framework and recommendations are clearly presented and thoroughly explained within the specific context, which is a great asset of the paper. The selection of references is relevant and recently published.

There are only a few segments to consider improving:

  1. In lines 61-64, the authors claim: “Besides, this article provides English teachers in Indonesia with practical recommendations for implementing EFL education for sustainable development through the integration Information and Communication Technology (ICT).” Although the recommendations are beneficial to English teachers in Indonesia, here, as well as in some other places in the paper, it should be added that other contextually similar teaching environments can also benefit from it.

Response 1: Thank you very much for recognising the paper and for your insightful comments. The authors agree with your feedback. Consequently, the authors have revised the sentences (currently in line 67-70)

“Besides, this article provides practical recommendations for implementing EFL instruction for sustainable development by incorporating Information and Communication Technology (ICT) for English teachers in Indonesia and other countries with a similar sociocultural context.

Point 2: In lines 88-92, the end of citation from the reference 12 is clear, but the beginning of the citation is unknown.

Response 2: Thank you very much for pointing this issue. We have double checked the reference and deleted the quotation mark at the end of the sentence to make it clear for the citation. The whole sentence (currently in lines 98-107) refers to reference 12. We are sorry for the confusion.

Point 3: In line 440, the tertiary level is related to the country (Indonesia) not the language (Indonesian).

Response 3: Thank you very much for your detailed review. We have revised this part, changing the word ‘Indonesian’ into ‘Indonesia’ (currently in line 618)

Point 4: Although the English language and grammar is correct, the paper could benefit from the proofreading by an expert, since a number of sentences do not sound as English natural.

Response 4: Thank you very much for your valuable suggestion. The professional proofread is kindly done to improve the quality of the manuscript.

Again, we highly value all of the constructive feedback on our manuscript. If additional modifications or information are required, we are happy to consider and make revisions. Thank you once more for all the positive feedback and helpful suggestions for enhancing the quality of this manuscript.

Best regards,

The authors

Reviewer 5 Report

The connections you've made to EFL of the principles of Sustainable Development Goals and Blended Learning are potentially of great value for the design and implementation of effective teaching.

Author Response

Response to Reviewer 5 Comments

Point 1: The connections you've made to EFL of the principles of Sustainable Development Goals and Blended Learning are potentially of great value for the design and implementation of effective teaching.

Response 1: We greatly appreciate your recognition and sincerely hope that our article can assist EFL practitioners in incorporating the sustainable development goals by implementing blended learning.

Again, we highly value all of the constructive feedback on our manuscript. If additional modifications or information are required, we are happy to consider and make revisions. Thank you once more for all the positive feedback and helpful suggestions for enhancing the quality of this manuscript.

Best regards,

The authors

Round 2

Reviewer 1 Report

The authors have revised the paper according to the research type. 

The paper can be considered for publication. 

All the best.